# Experiences of Faith-Based Organizations as Key Stakeholders in Policy Responses to Human Trafficking

**Charles Hounmenou** 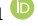

Jane Addams College of Social Work, University of Illinois Chicago, Chicago, IL 60607, USA; chounm2@uic.edu

**Abstract:** Faith-based organizations (FBOs) are substantially involved in the anti-human trafficking movement. Yet, limited research is available on their crucial roles in the field. This study explored their input in anti-trafficking policy implementation in the US by examining their motivations to engage in anti-human trafficking work, their distinctive competencies as stakeholders, and their experiences and challenges in providing anti-human trafficking services. A purposive sample of 16 leaders from 14 FBOs with anti-human trafficking work experience was recruited. A semi-structured interview guide was used to collect data. A thematic analysis of the data was conducted. The findings showed that FBOs have experience in various aspects of prevention, protection, and even assistance in prosecuting human trafficking cases and at multiple levels of intervention. The distinctive capacities of FBOs for policy advocacy, training, and housing services for trafficking survivors provide a glimpse of their leading roles in human trafficking policy implementation. Operating primarily outside public funding allows FBOs to develop short-term and long-term services for trafficking survivors without time constraints. The FBOs in the study reported using a non-discriminatory, survivor-centered, and trauma-informed approach in their anti-human trafficking service delivery. All the respondents in the study concurred that efforts by any FBOs to convert trafficking survivors to a particular faith are unethical and counterproductive. The implications for practice, policy implementation, and research are discussed.

**Keywords:** religion; faith-based organization; human trafficking; stakeholder; service; policy; experience



## 1. Introduction

Faith-based organizations (FBOs) are key stakeholders in the fight against human trafficking [1,2]. They have had a historically prominent role in anti-trafficking efforts in Europe and the United States. Christian missionary societies established the anti-trafficking movement in the 1900s [3]. Likewise, they strongly influenced the revival of the anti-trafficking discourse in the 1970s [4,5]. They were engaged in the fight against human trafficking long before the UN's Protocol to Prevent, Suppress, and Punish Trafficking in Persons of 2000 (Palermo Protocol) [5,6]. Yet, a lack of knowledge about why and how they do anti-human trafficking work hampers their legitimacy in the field.

An FBO is a non-governmental organization (NGO) established based on a faith tradition from which it derives guidance and impetus for its work in communities [6–8]. In other words, an FBO is an organization with ties to a religious institution and an underpinning faith ethos [8]. According to John-Michael [9], FBOs can better be described along a continuum, with at one end, those aiming to expand their faith with charitable services as a strategy to access communities through their religious messages, and on the other end, those compelled by their faith to protect oppressed groups and provide services without any expectation of conversion of service recipients.

### 1.1. Faith as FBOs' Major Inspiration for Anti-Human Trafficking Work

FBOs' engagement in the fight against human trafficking is justified in the sacred texts of major religions, including Christianism, Islam, Judaism, Hinduism, Buddhism, and their various branches [1,10–12]. Sacred texts such as the Torah, the Bible (including the Old Testament and New Testament), the Koran, and the Vedas, etc., contain sections calling on believers to fight for and protect the poor, the oppressed, and the weak [10,12,13]. In a review of the influence of sacred scriptures in the fight against human trafficking in the US, Barrows [10] highlights the New Testament's rationale for Christian FBOs' passion, the Torah's influence on Jewish FBOs' drive, the Koran's effect on Muslim FBOs' motivation, and the Bahá'u'lláh's inspiration in Bahá'í FBOs activism for the fight against human trafficking. For instance, the Torah commands Jews to protect strangers and demonstrate empathy for the most vulnerable [13]. Jewish FBOs' impetus for anti-human trafficking work has been connected with the Jewish people's historic journey to freedom [13]. Likewise, UNODC's report titled *Combating Trafficking in Persons in Accordance with the Principles of Islamic Law* [12] stresses that Islam prohibits labor and sexual exploitation. Many Christian anti-trafficking FBOs in the US identified their sacred scriptures as the driving force behind their work [11]. Those in the anti-trafficking movement in the UK identified William Wilberforce, a prominent Christian British abolitionist who crusaded to end the slave trade in the British Empire, as their pioneering source of inspiration for their work [11].

FBOs have substantially influenced the development of human trafficking policies in the US and the UK [1,6,8,14]. FBOs of Christian denominations have considerably influenced how human trafficking has been conceptualized in the USA [5,10]. For instance, Evangelical Christian FBOs were instrumental in developing the seminal human trafficking legislation, the Trafficking Victims Protection Act (TVPA) of 2000 [5,15]. The Washington Inter-Religious Staff Community Working Group on Human Trafficking's [16] report shows that faith has a crucial influence on the engagement of major FBOs in the fight against human trafficking in the USA. The Wilberforce Trafficking Victims Protection Reauthorization Act of 2008 is a salient illustration of the influence of Wilberforce-like Christian activists in the US anti-trafficking policy responses. They are prominent in anti-human trafficking service provision, campaigning, and political lobbying. In 2014, global leaders from the Christian, Buddhist, Hindu, Jewish, and Muslim faiths met in Vatican City to sign the *Joint Declaration of Religious Leaders Against Modern Slavery* [17]. The Joint Declaration conveys a unified faith-based proclamation that human trafficking is intolerable [5]. Faith-based communities, organizations, and congregations are powerful and important forces to count on in the fight against human trafficking [18]. According to Harrelson [6], FBOs' input in the anti-trafficking movement is likely to increase, which makes them influential stakeholders, strong adversaries, or powerful allies for other stakeholders. This author argued that FBOs have some leverage in doing anti-trafficking work because of their religious sensibility, and their motivation to often go above and beyond the call of duty and offer help in ways that enthuse an unusual level of trust among the people they serve.

FBOs' motivation for anti-human trafficking work is perceived as narrow because many tend to focus primarily on sex trafficking [11,19,20]. Weitzer [19] argued that Christian FBOs use their fight against sex trafficking to push their anti-sex policy agendas. FBOs sometimes use their services to convert human trafficking victims to their faiths [11,19,21]. Likewise, it was found that some religious leaders would utilize faith to coerce individuals and families into situations of exploitation [11,22,23]. Traffickers too would use religious language to ensure that trafficking victims are not able to understand their situation of exploitation [22,24]. In a review of three court cases, Heil [22] discussed how church leaders used religious beliefs to control members and maintain them in situations of involuntary servitude. In the case of the United States v. Lewis (1986), members of the House of Judah, a black religious sect, especially children, were routinely whipped for failing to complete forced work orders, including cleaning barns, cutting grass, caring for cattle, and cleaning toilets. The successful prosecution of the case was partly due to a link established between the religious leader's assumed biblical authority and the result of evident abuse,

exploitation, and seclusion. Another criticism researchers such as Harrelson [6] have against FBOs is that they sometimes exclude people who do not share their religious views or they try to convert them, which raises an ethical issue. Consequently, Harrelson [6] argued that FBOs could not lead the fight against human trafficking unless they prioritized assistance services over proselytizing.

However, some researchers (e.g., Green and Sherman [25]; Lewis et al. [8]; Twombly [26]) do not share the perception that FBOs prioritize proselytizing over assistance services. Green and Sherman [25] found that only 20% of FBOs regularly asked the clients they served to participate in religious activities. Twombly [26] found that faith was not prominently emphasized in the services provided by many FBOs, even though the spiritual nature of their work was useful and distinctive. In their recent three-year national study in the UK, Lewis et al. [8] found almost no evidence that FBOs involved in the anti-trafficking service field "were riddled with direct evangelism, proselytism, and spiritual abuse... despite some fear that this is the case", (p. 3). The FBOs in that study were found to have strong positions against proselytization [8]. Lewis et al. found that anecdotes of proselytism and spiritual abuse in services to trafficking victims were often associated with very few FBOs.

*1.2. FBOs' Engagement in the Fight against Human Trafficking*

FBOs play important roles in the areas of the prevention of human trafficking and the protection of the victims [1,10,26–30]. They can access substantial resources through religious networks [6,31]. Faith traditions and established social networks in some communities allow FBOs to sustain their anti-human trafficking work [6,30]. Lewis et al. [8] found that FBOs represent around 30% of analyzed service responses to human trafficking in the UK. FBOs in the US are engaged in anti-human trafficking work at various levels, including community, state, or national (e.g., Salvation Army; My Project USA; T'ruah; Hope for Life; etc.) [10]. They intervene in human trafficking prevention through advocacy and awareness-raising at the local, state, and federal levels [30]. Barrows [10] reported that the FBO called My Project USA was involved in community-level human trafficking prevention through its Muslims Against Human Trafficking initiative, which started in 2020 in Columbus, Ohio, after the arrest of 29 Muslim gang members who had trafficked minor Muslim girls for 10 years. T'ruah has created online educational resources to empower US Jewish communities to engage efficiently in various anti-trafficking efforts at a local level [10]. Another FBO, the National Council of Jewish Women, trains its members to use traditional and social media to raise awareness about sex trafficking through its Exodus initiative [10]. Some FBOs focus on specific forms of trafficking (e.g., sex trafficking or male trafficking, etc.) and categories of victims (children, males, or females, etc.) to be effective in their services (e.g., Emmaus focuses on male sexual exploitation) [10]. Other FBOs are engaged in changing policies affecting human trafficking through legal advocacy at the local, state, and federal levels. For instance, Shared Hope International focuses on domestic minor sex trafficking (DMST) policies at the state and federal levels (10). Similarly, Lewis et al. [8] found that FBOs in the UK are more likely to be single-issue trafficking organizations delivering more direct services than their secular counterparts; they also significantly contribute to political lobbying. FBOs also assist law enforcement agencies in investigations of human trafficking cases. For example, Hope for Justice USA, through its Tennessee Investigative Center made of private investigators, helps several law enforcement agencies with investigations of trafficking cases in Tennessee and beyond [10].

*1.3. Contribution to the Literature on FBOs' Anti-Human Trafficking Work*

Much of the limited research on FBOs' anti-human trafficking work is conceptual. It consists of systematic reviews of organizational reports and websites or grey literature reviews. Reviews of the literature on FBOs regarding services other than human trafficking show that evaluations of FBOs' services were basic and lacked reliability [10,32]. Lewis et al. [8] arguably conducted the most extensive, empirical research on FBOs' practices and activities in response to human trafficking in the UK, primarily in England. They

found that FBOs provide access to alternative resources, as they benefit from non-restricted private funding, which allows them to offer critical services and longer-term support. Lewis et al. [8] argued that "Access to unrestricted funds also provides FBOs with some freedom from governments or funders' whims about the extent to which modern slavery is a funding priority," (p. 23). By filling the gaps in services and relying on private funding, FBOs might inadvertently prop up public underfunding for human trafficking.

Similar to Lewis et al.'s research on FBOs' practices and activities in services to human trafficking in the UK, it is important to explore FBOs' roles as key stakeholders in the fight against human trafficking in other developed countries, so as to better understand how they contribute to human trafficking policy responses in diverse contexts. The present study examines the motivations, competencies, experiences, and challenges of FBOs involved in anti-human trafficking work in the American context. Four research questions are addressed to achieve this goal: (1) What factors motivate FBOs' engagement in anti-human trafficking work? (2) What competencies make FBOs key stakeholders in the fight against human trafficking? (3) What are FBOs' experiences, contributions, and challenges in anti-human trafficking work? And (4) How can FBOs' input in the fight against human trafficking be enhanced?

## 2. Methods

### 2.1. Research Design and Participant Selection

A qualitative research design, with an interview method, was used. A purposive sample of participants from FBOs with experience in anti-human trafficking work in the United States was selected based on two criteria: (1) be from an FBO with at least five years of experience in anti-human trafficking services, and (2) be designated as a spokesperson regarding the organization's anti-human trafficking work. Three strategies were utilized to identify organizations from which to recruit participants in the US. First, a Boolean search was conducted with a combination of four groups of keywords: (a) religion, religious, spiritual, faith, faith-based, order, organization, center, services, and association; (b) human trafficking, trafficking in persons, exploitation, sex trafficking, labor trafficking, servitude, sexual exploitation, slavery, and sex trade; (c) Christian, Evangelical Christian, Protestant, Roman Catholic, Jewish, Baha'i, Muslim, Buddhist, Hinduist, and traditional faith; and (d) drop-in, shelter, safe house, rescue, services, victim services, assistance, outreach, campaign, rehabilitation, reintegration, support, and rescue. Second, a literature review on faith and human trafficking was conducted to identify more FBOs. Third, the information on the websites of the identified FBOs was reviewed to assess whether the organizations were still involved in anti-human trafficking work and then to collect their contact information (i.e., e-mail and phone numbers).

Twenty-five FBOs from the following faith denominations were identified based on the criteria mentioned above: Christian (i.e., Evangelical Christian, Protestant, and Roman Catholic, etc.), Jewish, Baha'i, Hinduist, and Muslim. The leaders of the organizations were contacted by e-mail and/or phone to inform them about the research project, request that they designate one spokesperson knowledgeable about their organization's anti-human trafficking work experience, and send the latter's contact information for recruitment into the study. Eleven of the 25 FBOs did not participate in the study for several reasons, including failure of organizations to respond to invitations or identify a spokesperson, refusal of spokespersons to participate in the study, lack of responses from spokespersons, time conflicts for interview scheduling, or organization turnover. The 11 FBOs were from the following faith denominations: Muslim (2), Baha'i (1), Hinduist (1), Jewish (3), and Christian (4). Fourteen FBOs of five faith denominations (i.e., Roman Catholic, non-denominational Christian, Evangelical Christian, Protestant, and Jewish) participated in the study. They were each represented by one participant, except for two organizations that requested having two participants because it was believed that just one participant would be unable to respond to all the questions in the interview guide, which was shared with

the organizations days before each interview. Thus, 16 respondents, representing 14 FBOs, participated in the study.

A semi-structured interview guide was used to collect data from each of the 16 participants via Zoom. The interview guide included questions exploring the following variables: (1) characteristics of the participants and their FBOs; (2) motivations of the FBOs to engage in anti-human trafficking work; (3) competencies and resources of the FBOs involved in the fight against human trafficking; (4) experiences, contributions, and challenges of the FBOs in their anti-anti-human trafficking work; and (5) ways to enhance the FBO's input in the fight against human trafficking. The average time for each interview was 45 min. The institutional review board at the researcher's university approved the study. All the respondents gave informed consent to participate in the study. To protect the privacy of participants and their organizations, pseudonyms are used in the data analysis.

### 2.2. Data Analysis

A thematic analysis of the interview data was conducted. Thematic analysis is a method for identifying, analyzing, and reporting patterns or themes within a qualitative dataset. It is used to minimally organize and interpret various aspects of the research topic. It is an appropriate method to examine and understand experiences, thoughts, or behaviors across a dataset [33]. The transcribed interview Word files were reviewed and categorized into major sections. The Atlas.ti 8 qualitative data analysis program was used to code and analyze the data. Ten of the 16 respondents were sent the analyzed data for member checking, that is, to check for the accuracy of the analysis of the information they provided, and the interpretation made.

### 3. Findings

### 3.1. Participant Organizations' Characteristics

Table 1 displays descriptive statistics about participants' positions in their organizations and the organizations' geographical scope of anti-human trafficking work, faith denomination, and length of time in anti-human trafficking work. Thirteen of the 14 FBOs in the study were of several Christian denominations (93%), and one organization was of a Jewish faith denomination (7%). All the participant organizations provided services at multiple geographic levels (county, state, federal, and/or international). All 16 participants were in leadership positions in their respective organizations. Seven organizations (50%) had provided human trafficking-related services for 4–10 years, while six (43%) had done so for 16–32 years by the time the study was conducted. While seven of the 14 FBOs primarily dealt with female sex trafficking, one dealt with male sex trafficking, and six with both sex trafficking and labor trafficking. Each FBO provided more than one type of human trafficking-related service. Outreach to trafficking victims and awareness-raising were reported as key areas of work by more than half of the FBOs. Ten organizations provided training services about human trafficking, whereas three FBOs primarily provided safe housing services, and three others provided services on policy advocacy.

Research was part of the human trafficking-related activities of two agencies. Four FBOs reported being among the largest ones nationally in the service areas of housing, training, and advocacy about human trafficking. One FBO reported being a national leader in shelter services in the USA. Another FBO, consisting of business investors and shareholders of Christian denominations, could be arguably considered one of the rare, major ones in the USA advocating against human trafficking within large corporate companies.

**Table 1.** Demographics of respondents and their organizations.

| Variables | Frequency | Frequency |
|:---:|:---:|:---:|
| Participant's position in organization (*n* = 16) | | |
| Executive Director | 5 | 31.3 |
| Program Director | 5 | 31.3 |
| Committee Leader | 2 | 12.2 |
| Chief Operations Officer | 1 | 6.3 |
| Board member | 1 | 6.3 |
| Community Outreach Director | 1 | 6.3 |
| Vice President | 1 | 6.3 |
| Total | 16 | 100.0 |
| Organization's geographical scope of anti-human trafficking work * (*n* = 14) | | |
| Citywide/countywide | 6 | 32.0 |
| Statewide | 3 | 16.0 |
| Nationwide | 8 | 42.0 |
| International | 2 | 10.0 |
| Organization's faith denomination (n = 14) | | |
| Roman Catholic | 5 | 35.8 |
| Evangelical Christian | 4 | 28.6 |
| Non-denominational Christian | 3 | 21.4 |
| Protestant Christian | 1 | 7.1 |
| Jewish | 1 | 7.1 |
| Total | 14 | 100% |
| Organization's length of time in anti-human trafficking work (*n* = 14) | | |
| Four years | 1 | 7.1 |
| Six years | 1 | 7.1 |
| Seven years | 1 | 7.1 |
| Eight years | 3 | 21.3 |
| Ten years | 1 | 7.1 |
| 16 years | 2 | 14.2 |
| 20 years | 2 | 14.2 |
| 24 years | 1 | 7.1 |
| 32 years | 1 | 7.1 |
| Over 40 years | 1 | 7.1 |
| Total | 14 | 100.0 |

* Noncumulative statistics.

### 3.2. FBOs' Motivations for Engaging in Anti-Human Trafficking Work

Two key themes were highlighted in the FBOs' rationale for dealing with human trafficking, including primary and secondary motivating factors.

### 3.2.1. Faith as a Primary Motivating Factor

Faith was reported by all 16 respondents as the overriding, connecting factor that substantiated the involvement of their FBOs in the anti-trafficking movement. The findings show an overwhelming influence of faith not only over the impetus of FBOs for anti-human

trafficking services but also in their service delivery approach. Subthemes that indicate the influence of faith on the FBOs' anti-human trafficking work include the following: faith as a driving philosophy; opposition of all religions to slavery; anti-human trafficking work as an appropriate field for practicing faith; and connection between human trafficking and history of persecution.

Faith as a Driving Philosophy

The findings show that faith pervades all parts of FBOs' anti-human trafficking activities. Isaac, the leader of the only FBO addressing male trafficking, said, "It is our faith that motivates us to help. . . . If someone is a believer, it is innate within them to help. I think when we love our neighbors as ourselves, we see their humanity, and we see their needs; therefore, we are compelled to serve." Faith calls for FBOs to prioritize the life and dignity of victims. Annie explained, "Most faith traditions have an idea of basic human dignity. That is why they come to this work naturally." Even though there is a variety of faiths, all the participant FBOs share the core value of religious teachings about assisting vulnerable groups. As Lisa observed:

> All Christian faiths are motivated by the teaching of 'love thy neighbor as thyself.' But we have dealt with Jewish organizations, Muslim organizations, or even individual members of other organizations of different faiths, and they have similar teachings. So, when we work with them, we share certain philosophies.

Nicole, a respondent from a national policy advocacy FBO, explained that it was thanks to their Christian faith that the founders of her agency embraced anti-human trafficking work. They were firm believers and felt their responsibility was to help victims without discrimination.

Yet, people have difficulty understanding how faith impacts the passion of FBOs to dedicate time and resources to a dangerous field such as human trafficking. Thus, they are confused when respondents such as Jean and Amanda quit lucrative businesses or positions to establish FBOs dedicated to anti-human trafficking work. Justifying her faith-laden passion, Jean said:

> One of the things that we bring to people who are not people of faith is a mystery about what compels us. I worked for just over 16 years in business and left. Thus, thousands of people will go, "I don't get it. Why would you do this? Why would you give up? You were making a lot of money in Washington, DC. Why would you do that?" I think the mystery of why people of faith are compelled to do extraordinary things is initially discordant, but over time it's very attractive.

Mandated by Faith to Fight Human Trafficking

All 16 respondents concurred that all major religions oppose slavery. Jenna, a regional leader of one of the biggest international FBOs, explained, "When you look at the faith community, human trafficking goes against the values of all major religions." She was among 12 participants who reported that the sacred scriptures of most religions mandate loving, serving, and supporting any people needing help. Confirming the same point, Nicole explained that faith justified the establishment of her agency because human trafficking provides a unique opportunity to demonstrate it:

> Our staff are all faith-centered, and our volunteers align with our mission and values. What makes that unique is that we are bringing our faith into the foundation of who we are, through community engagement and people in the community.

Sara, the respondent from the only Jewish FBO in the study, observed that human trafficking was comparable to the persecution the Jewish people experienced for centuries. She said that Jewish groups were pioneers in the anti-trafficking movement in the United States:

> There are just so many times over history, including biblical history, that the Jews stood up against injustice and the value of human life and the value of

human dignity. Our trafficking program is part of an organization that was at Ellis Island over 125 years ago, making sure that young girls weren't taken by potential traffickers and put into child labor or sex trafficking.

Five respondents believed faith helps FBO workers cope with the vicarious trauma and burnout they experience in working with trafficking victims.

> One key attribute of the faith-based community is that when you set your aspirational goal higher than the success of one individual, you are able to survive the high fluctuations, the amount of turnover, the trauma, and the violence of the stories you have heard from victims. (Jean)

They also argued that faith provides an environment to empower victims for dignity and self-determination:

> The edict of love oversees all. We're still going to love people, whatever their choices are. For us, it's quite an easy decision that love oversees everything. So, we don't have to make decisions to discriminate or to judge because we're just supposed to love people. (Amanda)

There is self-determination when someone sets their own goals and works to achieve their goals [34]. It contributes to positive results in areas that affect a person's life decisions and well-being.

### 3.2.2. Secondary Motivating Factors

Four secondary motivating factors were identified from the data, including personal experience or witnessing human trafficking, the need to fill the gaps in victim services, the participation in educational events, and the call for social justice.

#### Personal Experience or Witnessing Human Trafficking

Five respondents created or joined their current organizations after seeing the exploitative situations of trafficking survivors or former clients. Describing how she decided to quit a lucrative corporate business position to establish her FBO, Jean stated:

> Human trafficking survivors have so changed me and shaped my identity and my understanding of the problem that I would say that that has given me a tenacious, unrelenting pursuit because all of those people are now a part of what drives me to do that.

Amanda, a survivor of sex trafficking, who also used to own a business company and then became the leader of a major FBO providing safe housing and training, was inspired by an encounter with an exploited person on the streets of Baltimore. She said, "That interaction compelled me to sell my company and my home and buy a property to start the first shelter program for victims in the mid-Atlantic area." The founder of another FBO, a former US Congresswoman, established her organization that focused on anti-trafficking policy advocacy after witnessing sex trafficking in Asia.

#### Addressing the Victim Services Deficit

In total, 9 of the 14 FBOs were established to address the shortage of human trafficking services and resources in their communities. The following subthemes were highlighted about gaps in anti-human trafficking services and resources in their communities: feeling called to address the deficit in victim services; gaps in housing for sex trafficking victims; assisting victims unable to receive services anywhere; and assisting law enforcement in dealing with sex trafficking.

Four FBOs decided to provide services for human trafficking victims traditionally disregarded by secular organizations and public agencies. Isaac said he felt compelled to create his organization to fill the gap in basic services for male sex trafficking survivors because no agencies wanted to assist this population in his state. Amanda's FBO provided

services to a unique group of survivors unable to receive assistance anywhere, that is, sex-trafficked women with kids and pets. As she explained:

> A service gap that we saw was women with children and pets. So now we pretty much only serve women with children and pets. Essentially we've become the organization that you send people to if they do not fit anywhere else. There are actually no other places that accept trafficking survivors with pets. So, we have, over the years, developed into filling this niche.

Respondents reported that law enforcement agencies would contact their FBOs to help prevent the sex trade in their districts and provide support during or after rescue operations. Participation in educational and awareness-raising events about human trafficking was also reported as a major prompt for establishing 7 of the 14 FBOs.

Call for Community Engagement and Social Justice

Some FBOs decided to engage in the anti-human trafficking movement to address an apparent detachment of people in their communities toward the problem. Describing her knowledge about how Jewish communities gradually became involved in anti-human trafficking work, Sara explained that:

> The myth that it could never happen to anyone Jewish or that there are no Jewish traffickers was just something that we sought to dispel. It is a major problem for the world, so we should be part of helping to address it. It just made sense that we would raise our voices.

Sometimes, FBOs are established because faith community members believe that doing something about human trafficking is a tangible way for their congregations to demonstrate their mission to fight for oppressed groups.

The call for social justice was another key factor that motivated FBOs' involvement in the anti-human trafficking movement. This call was reflected in the following elements highlighted from the data: passion for social justice; fighting for the voiceless; compassion for the oppressed; and call for fair labor practices in for-profit companies. All the respondents reported social justice as one of the core values sustaining their FBOs' anti-huma trafficking work because every religious tradition has a philosophy such as 'Do unto others as you would have them do unto you.' Amanda pithily summarized the shared view of all participants about social justice as an impetus for faith communities to fight human trafficking:

> Are there any religions in the world that don't believe in social justice? I don't think so. I haven't come across any. I believe that that is an edict within every religion, whatever it is. Whether you're a Jew, whether you're a Christian, or whether you're a Hindu, we all believe the same when it comes to justice. Nobody believes that someone should be treated inhumanely or injustice. The issue of human trafficking has that built into it from start to finish. It is entirely injustice. All religions believe justice and equality, and equity are necessary for people to feel part of their community.

*3.3. FBOs' Competencies for Anti-Human Trafficking Work*

Respondents' perceptions of FBOs' competencies for anti-human trafficking work highlight several subthemes in areas of prevention, protection, and prosecution.

3.3.1. Prevention-Related Competencies

Subthemes that emerge from participants' perceptions about FBOs' competencies for human trafficking prevention activities include: capacity for accessing various community resources, ability for awareness-raising and outreach, ability for policy advocacy, reliance on volunteers, and fundraising capacity.

Six respondents concurred that FBOs could tap into various congregation and community resources for human trafficking prevention work. They could mobilize social support

within their faith communities for awareness-raising and outreach campaigns. For Amanda, "They [FBOs] are basically using their community resources to do preventative work. When you get a community behind something, it creates momentum that moves forward." In the same way, Jenna said, "They have a lot of safety nets in place, not only in intervention but also in prevention work. Because these networks and resources are already pretty well established, many potential trafficking victims are already actively engaging with FBOs and faith-based communities." FBOs' ability for grassroots organizing is critical for their advocacy and outreach for human trafficking. They often provide street and night outreach services to sex trafficking victims. They are uniquely positioned to reach survivors and their families through their connection to church congregations:

> If survivors can find their churches, either as a safe haven or if they can reach out to those who might be able to reach others in their community, it is really the grassroots way of getting back into the community to address these issues. (Megan)

Five respondents perceived FBOs to be very influential in outreach and awareness-raising and in the sphere where policymaking for human trafficking occurs. FBOs can reach out and mobilize communities for policies that address human trafficking, especially sex trafficking and the related problem of prostitution. Jean claimed that faith communities were influential in successfully passing major trafficking laws in the last two decades in the US, starting with the Trafficking Victims Protection Act of 2000.

3.3.2. Protection-Related Competencies

The findings highlight five subthemes on FBOs' ability to provide protection-related care to trafficking survivors, including: being active in victim services; the capacity to provide survivor-led trafficking services; providing safe spaces for trust-building; the ability to deliver long-term recovery and reintegration services; and the dual worth of FBOs' input in the fight against trafficking.

The findings highlight the capacity of FBOs to provide a space for survivors to develop trust and use their self-determination in services they need. Based on the services they receive and how service providers treat them, survivors can determine if the therapeutic relationship is safe and unconditioned. As Annie explained:

> I think survivors of human trafficking can wonder, "Why are these people helping me? Why would they even do that?" If a survivor comes into a situation, they may be a little leery of anyone, and rightfully so, and if they can make a reasonable leap in their head as to why someone is helping them, they are more likely, I think, to trust them.

FBOs' distinctive ability to serve human trafficking survivors could be explained by the fact that some of their staff have personal stories of exploitation, which allows them to show compassion while encouraging survivors' self-determination. "From my experience, I see that most people who join these faith-based organizations have a testimony. They all have some type of trauma or whatever the situation was," explained Julie. Five respondents stressed the importance of considering trafficking survivors' input in the services they receive. They reported that faith calls for self-determination in the choices survivors make. Amanda explained that:

> If and how are they going to make a choice to request spiritual guidance? God gives us free will. If we live out our faith, if we live out our love, if we do what we're supposed to do as humans, then naturally people will get drawn to ask, "What is that thing within you that gives you the ability to love this much, that gives you the ability to just forgive this much?" And when they start asking, then we can start helping them.

Thus, opportunities for spiritual development are offered to trafficking survivors when they seek such information. As Isaac stated:

I think that our teaching is that people need to make their own decisions. It is totally up to survivors if they want to participate in a religious program. Or if they don't want to participate in it, they still have access to every program and resource that we offer. We believe that if they do, they may have their reasons for it. But it's not a prerequisite at all exactly. It's our responsibility to just love and provide support to folks.

FBOs' active role in victim services is highlighted through three interconnected sub-themes: the ability to provide long-term services; using an approach like Maslow's Hierarchy of Needs; and unlimited service time for survivors. Five respondents argued that time-constrained services could impede trafficking survivors' chances to recover from their trauma. Considering that it takes time for traumatized survivors to recover from abuse, some FBOs would use the approach of Maslow's Hierarchy of Needs in their services to trafficking survivors, allowing the latter to reach a level of comfort where they feel safe to reach out for other needs.

We believe in Maslow's Hierarchy of Needs in which victims' needs need to be met before we consider their spiritual needs paramount. So, before we can worry about their spiritual needs, we need to worry about their physical needs such as water, shelter, clothing, psychological safety, and the very basic needs that have to be met. Over time, they will see us living out our faith and asking about it, but it's not for us to tell them and force it upon them. It's for us to show them what love looks like. (Amanda)

Maslow's Hierarchy of Needs depicts five levels of human needs [35]. The first level concerns physiological needs (e.g., food, drink, shelter, clothing, and sleep), considered the most important survival needs. The other four levels depend on meeting these first-level needs. The second-level needs are safety needs (e.g., protection, safety, law, well-being, stability, and freedom from fear). Social needs—the third-level needs—involve being part of a group, belongingness, trust, acceptance, and relationship. The fourth level is self-esteem (dignity, achievement, independence, and industry) and the desire for respect. The final group of needs—self-actualization—is about achieving personal potential, self-fulfillment, seeking personal development, and desiring to thrive and achieve everything one is able to [35].

The importance of FBOs' input in the implementation of anti-human trafficking policies could be explained through both theoretical and practical perspectives, as perceived by two respondents. Through compassion and guided by faith, FBOs instill in survivors hope for restoration; more importantly, they fill the gaps in anti-trafficking-related services that are rarely accounted for in policy evaluations. Jean, one respondent, summarized this view:

First, on a theoretical plan, one of the distinctive characteristics of FBOs is that we're predicated on hope. If you think of harm reduction as a philosophy, it says people can just stay in addiction, but we want it to be safer; there's nothing aspirational or hopeful about that; there's nothing in there that says, "life could be better." In the mix of options, trafficking survivors need to see that there's hope. Second, most of these faith-based agencies take no government funding. So, if you want to be pragmatic, that's a state or a nation's cheapest option. That's just a practical reality because somebody else is paying for all of that care. FBOs are very connected to their local community, very embedded in cultures and local resources, and are able to do things that the top-down approach can't get done. So FBOs in this work are providing a valuable option.

FBOs are greatly represented among victim service providers, providing shelter services, housing, drop-in center services, and mental health services, etc. According to Jean, "The faith-based community is probably the most robust and active in the victim services arena. Where you would see them most obviously and perhaps in the larger numbers is in victim services. And that's everything from running a drop-in center to street outreaches".

### 3.3.3. Support for Investigations and Prosecutions

Three respondents reported that law enforcement agencies would contact FBOs for help during investigations. They argued that assistance to law enforcement could improve the outcomes of investigations because FBOs could provide trafficking survivors with a sense of safety and trust while addressing their immediate needs, such as shelter and food. FBOs could help move survivors into a space other than detention, thus increasing the likelihood of the latter providing crucial information for investigations and prosecutions. Describing the crucial input FBOs could have in investigations, Jean observed:

> I've seen it work extraordinarily well when a law enforcement agency jurisdiction partners with faith-based organizations to provide the services that law enforcement can't do or that impede their progress. It looks like this: When law enforcement is going to do a sting operation, and they anticipate that they're going to apprehend thirty victims, they can't unless they charge those individuals, they can't legally hold them, but if they have a partner organization that can go on that sting operation with them to be that interface with the victim, not have the same agenda, not have the same authority, provide for victims' material needs, offer resources, their operation may be successful.

Two respondents reported that pimps sometimes condoned FBOs' help to trafficking victims and even asked for spiritual guidance. They tolerate FBO staff because the latter are not law enforcement. Julie observed, "There have been times when because we are faith-based, the pimps fear us. Once they hear we're from the church, they'll be like, 'Oh, okay.' And as crazy as it sounds, sometimes the pimps ask and receive prayer more than the women." The non-threatening presence of FBOs in a trafficking environment highlights their ability to connect with victims and traffickers.

### 3.4. Experiences and Contributions of Participant FBOs

The 14 FBOs' anti-human trafficking work experiences centered on the following service categories by frequency: training services ($n = 10$); awareness-raising and outreach campaigns ($n = 9$); victim services ($n = 4$); safe housing ($n = 3$); policy advocacy ($n = 3$); research ($n = 2$); and funding support ($n = 1$). The findings on the 14 FBOs' experiences and contributions highlight three subthemes: salient aspects of services based on the 3 "Ps" approach of prevention, protection, and prosecution; ethical considerations in the services provided; and funding resources for anti-human trafficking work.

### 3.4.1. Salient Aspects of Participant FBOs' Trafficking-Focused Services
Prevention Efforts

Five subthemes are highlighted in the prevention-focused trafficking services that the 14 FBOs provided, including awareness-raising and outreach; training; policy advocacy; research; and capitalizing on volunteers as an asset.

Education, Awareness-Raising, and Outreach Activities

Nine of the 14 FBOs reported conducting awareness-raising and outreach activities as part of their prevention efforts.

> At our church level, we organize speakers to speak at the masses. We got a dropdown on the church website to talk about our ministry against human trafficking. We had gatherings after masses so that we could introduce the idea of labor trafficking and how fair-trade ties into that. So, we were able to share a lot of those resources and do those things at our parish first. (Lisa)

Kathy reported that her FBO had a program of 1200 trained volunteers called Ambassadors of Hope, who were deployed to educate local communities about human trafficking and help them identify opportunities to engage with them. Over half of the FBOs in the study organized outreach campaigns that targeted mostly female victims in hotspots of sex trafficking or labor trafficking in major cities during social or sports events. Only one FBO

conducted awareness-raising activities targeting male survivors. This FBO was reported as one of only five national organizations that work with that population.

Internet and Communication Technology (ICT) was utilized by three of the FBOs in the study not only to track the moving of trafficking victims across cities and states by traffickers but also to reach out to victims to inform them about ways to request help. Amanda, an ICT expert and founder of her FBO, said:

> We do that to reach out to those locations specifically rather than just going to every massage place or strip club or whatever. We pinpoint the ones using artificial intelligence and machine learning and other things, to help pinpoint. So, we're not wasting our resources and also annoying businesses that are perfectly legal. And that way, we can target more people who are likely to be trafficked.

Likewise, Isaac's FBO used "Freedom Signal" software to monitor online transactions and reach out to potential male victims to inform them about specific resources available for this often-overlooked population. As he said:

> We utilize a technology called Freedom Signal, which actually has the ability to scrape websites where sex is being sold. Once we're able to get basic information, we can then send out text messages to people to say 'Hey, we are offering the service. If you know anyone who needs help, have them reach out.' We do it via technology and a lot of street outreach.

In raising awareness about human trafficking, respondents reported that FBOs enjoy high credibility with the public and communities they serve. Leah explained that thanks to her FBO's awareness-raising sessions, communities started realizing that human trafficking was a critical issue to be educated on. The success of those sessions could be justified by the trust people have in faith groups. As Leah said, "Awareness raising was our biggest strength, and the Sisters have much credibility because, first, of being Sisters, but second, they'd done a lot of social justice work in the community. Thus, people really trust them".

Training Services

Ten of the 14 FBOs provided human trafficking-related training services targeting diverse categories of stakeholders, including businesses, religious congregations and communities, schools, health and social service agencies, and other NGOs. Jean reported that her organization was one of the two national FBOs that provided training to sister organizations about trafficking survivors' access to care and quality of care.

> We have a training program for startups. So, if an NGO in the South Side of Chicago wants to start a shelter for victims of trafficking, they can go through our three-year program to shepherd them through the process of creating a non-profit, putting all of the infrastructure in place, learning about human trafficking and survivors, developing a care model enlarging their services. So that's probably our flagship program.

Megan's FBO specialized in trauma-informed training certification programs for stakeholders in the healthcare and education sectors:

> Our national prevention programs are a lot lately centered around training. We have an awful lot of education out there, both in terms of internet safety, in terms of training for healthcare providers, social workers, and emergency medical technicians. We have very extensive training in our justice initiatives.

Three of the FBOs implemented job training programs on behalf of survivors. Amanda's FBO taught survivors about budgeting, life skills, administrative skills, web design, and social media literacy, in its safe house.

Policy Advocacy

Policy advocacy was reported as one of the key service areas of three FBOs, including two national and one international. The data on these FBOs' policy advocacy work highlights faith-based groups' capacity to shape trafficking policies. Yet, the choice to focus on

policy advocacy often depended on FBO leaders' or founders' experience and expertise. That was the case with one FBO whose founder's lawmaking experience was crucial to her FBO's national expertise and influence on DMST policy reform.

One FBO could be considered one of its kind in addressing human trafficking from inside the corporate business system. Consisting of business owners and shareholders of Christian denominations, Daniel's FBO shows how faith-based business networking could be utilized to advocate for human trafficking policy changes in big companies and their supply chains. He reported that his FBO successfully caused major international companies, such as Marriott Hotels, to adopt an anti-trafficking policy that included training all their frontline staff at every hotel in identifying and preventing human trafficking. This FBO also advised other faith-based associations of investors about how to persuade corporate companies to address human trafficking in their businesses. It was reported as having played key roles in policies addressing human trafficking in a few US states and other countries. For instance, his FBO played a key part in Iowa's policy addressing trafficking in the hospitalization and hotel industries. Daniel's organization was instrumental in causing California's *Transparency in Supply Chains Act* of 2012 to be passed. It was instrumental in urging investors, NGOs, and other companies to support the UK Modern Slavery Act legislation.

Other policy advocacy activities on human trafficking that participant FBOs contributed to included organizing major conferences at state or federal levels, coordinating task forces or coalitions, and engaging in policy development at the state level. Discussing the leading role her FBO played in organizing national training conferences on human trafficking, Nicole said:

> A huge training opportunity is our JuST Conference—that is, the Juvenile Sex Trafficking conference–every year, which involves a thousand people that attend and go through multiple workshops. It's known to be the premier domestic minor sex trafficking training in the country.

Research Activities

Two of the 14 FBOs had research as a secondary anti-human trafficking activity. The research topics participant FBOs focused on include analysis of federal policies addressing the sex trafficking of children and women; care needs assessment; and forms of resources accessible to trafficking survivors; etc. As Jean said:

> From a mission perspective, we want to improve access to care and the quality of care. This is still a relatively young field; there's not much research, there's not much data; there are no standards. So, we want to help establish standards and help organizations meet those standards, if not exceed those standards, for the good of survivor care. Parallel to that, we continue to do a tremendous amount of research nationally, whether they are studies on changes in the victim population over time or key issues of care that we want to make decisions based on better data.

Capitalizing on Volunteers as an Asset

Nine of the 14 participant FBOs relied on volunteers with diverse expertise, which they considered a major strength of most FBOs. Volunteers were primarily congregation members, trafficking program graduates, and community members. Jenna described how her organization capitalized on volunteers, perceived as a great asset for many FBOs with limited access to funding for services to trafficking survivors:

> The way that our program has used volunteers is through a 24-h hotline. What is so great now is that we have developed a really incredible volunteer base where volunteers cover all of the shifts. So, we always have staff as a backup. We have volunteers who are able to engage in that kind of emergency triage phone call, a resource connection, helping people find emergency housing and all of that. And one of the things that we do well with the support of volunteers is that we have a

drop-in space for female-identified youth and young adults who've engaged in commercial sex.

Four of the leaders of FBOs in the study were trafficking survivors who founded their organizations or rose through the ranks to become agency leaders. Respondents believed relying on survivor-led teamwork was an excellent strategy for working with trafficking survivors. For instance, Amanda argued that having survivors at the core of their program explained most of her agency's successes:

> We have an amazing team of people, many of whom are survivors of sexual trauma or human trafficking, or domestic violence themselves. And so, because of that, they themselves have the resilience to them that helps us to serve our girls as fellow survivors. We have a *Moms Against Trafficking* that's run by a mom whose daughter was trafficked. We have a psychotherapist; his daughter was a victim of human trafficking. Our executive director was a victim of domestic violence. So, everybody's been through something. They understand what it's like to come out on the other side. And so, they have lived experience of what it takes to start over and make a difference.

Protection Efforts

Participant FBOs' service experiences focusing on protecting trafficking survivors are highlighted through the following subthemes: case management and referral services; survivor-centered and trauma-informed services; and housing and rehabilitation services.

Case management and referral services

Case management was one of the key services of 5 of the 14 FBOs in the study. While community-based case management service was a big part of the human trafficking-focused services provided by Jenna's FBO, humanitarian aid was one of the major services Lisa's organizations provided. Isaac's FBO provided comprehensive case management services to male survivors:

> We provide at our drop-in center a whole array of social services for male survivors of sexual exploitation. But more importantly, we provide counseling. We provide case management, and through that case management, we do an assessment. We do an action plan based on the assessment and based on the needs of that particular client. We will then make the necessary referrals for nine times out of ten housing, employment, and income assistance.

Isaac's FBO was distinctive because it provided case management services to male survivors, a population often overlooked in services to trafficking survivors. He said, "For people to know that there is an organization that works with males, that is uniquely different, and sets us apart".

All the respondents reported referral as an important aspect of human trafficking services provided by FBOs because trafficking survivors have too many needs for a single organization to address. Although Isaac's organization focused on male survivors of sexual exploitation, it helped both male victims of labor exploitation whenever possible and referred them for appropriate assistance. As he said:

> We don't turn anyone involved in labor trafficking away. If someone is in need or someone has been impacted by labor trafficking, we then have all the connections and resources to others in the community, where we would then provide referrals. Thus, if things are more specifically regarding labor trafficking, we would then refer out.

Housing and rehabilitation services

Housing services for trafficking survivors were reported as a major work area of 5 of the 14 FBOs. Describing her FBO's successful shelter program dedicated to survivors of trafficking nationwide, Jean said: "We have established ourselves as one of the best

resources for who's out there doing shelter work for survivors of trafficking." She claimed her FBO owned six of every ten shelters for trafficking victims in the US. As she explained:

Safe housing was the major service of three of the FBOs. Describing her FBO's long-term transitional safe housing program for sex trafficking survivors, Julie said:

> Our safe house program is a one-year program outside of the city, which is outside of the distractions, outside of all that memorabilia that they're able to see and what they've experienced. Survivors are able to get that recovery, restoration, and that renewed life. We've had over 150 women graduate from this program since the start. I myself am a 2019 graduate of this program, so I speak from experience of what this program can do.

Amanda's FBO's safe house program specialized in preserving family units by keeping women survivors together with their children and sometimes with their pets. Three FBOs with housing services reported having rehabilitation and reintegration services as part of their key areas of anti-human trafficking work.

> We have rehabilitation and reintegration programs that we offer, like *Ending the Game* and *Father Fracture*, and other classes that we offer. We use media to provide online therapies. They have a case manager assigned to them so that they basically get what we call a dream plan. And for every 90 days, they work through one or two of the things on the Arizona Self-Sufficiency to move them forward because our end goal is to take them to reintegration, which is independence. (Amanda)

Three respondents reported that their FBOs rarely put a time limit on services for survivors of trafficking because they liked using an approach similar to Maslow's Hierarchy of Needs. As described above, in this approach, it is important to start services by addressing survival needs, followed by safety needs, social needs, independence and dignity, and personal fulfillment. Discussing how her agency relied on this approach, Megan said:

> We like long-term restorative care because there's that continuing to meet individuals where they're at and not giving up. And a lot of times when you get federal grants, and there is a certain alignment, you have to get so much done within a certain amount of time. And you've got to prove numbers, whereas a lot of times with faith centers, it is the person over that process because they're worth the investment and the necessary time.

Input in Investigations and Prosecutions

Four of the FBOs reported having contributed to investigations or prosecutions of trafficking cases, mainly assisting victims during rescue operations by law enforcement. Not only did Lisa's FBO assist the police during rescue operations, but it also collaborated with major criminal justice stakeholder agencies such as the Department of Homeland Security and the Attorney General's Office in court cases in which trafficking survivors were key witnesses:

> We do several things with Homeland Security. For a raid in which they know that they're going to find human trafficking victims, they will contact us in advance and say, 'Do you have any backpacks? We're going to need those backpacks to get to these victims because when we pick them up, they're not going to have clothes or hygiene products or anything.' Probably the bigger assistance is when they are handling cases through the Attorney General's Office; they will prosecute those cases, usually in Los Angeles, but the victims will be kept in hotels in Long Beach or some other areas.

Her FBO would provide survivor witnesses with clothes that might be more appropriate to wear to court and even recreation event tickets because human trafficking trials could go on for weeks. As she explained, "We even gave survivors suitcases when they were leaving because they had accumulated things while they were here, but they had nothing

to take their things home. So, it is really anything on the victim assistance side of things." Her FBO also provided translation support services for survivor witnesses in court cases.

3.4.2. Ethical Considerations in Providing Services to Survivors

Ethical considerations highlighted in the findings about participant FBOs' experiences include using a non-discrimination policy in service provision, embracing survivors' self-determination in care, prioritizing survivor-centered, trauma-informed service approaches, and valuing diversity in care approaches.

Non-Discrimination in Service Provision

All the respondents indicated that their organizations had an anti-discriminatory policy, calling for everyone accessing their services to be treated with dignity and respect and without the consideration of faith. Faith should not be a condition for service access and benefits. Discussing her FBO's anti-discrimination policy, Jenna said, "My organization's international policy position statement says that we provide services without discrimination. That's not to say we never make mistakes. But we, as a Christian organization, should not be creating any barriers for folks." She explained that people seeking services and support from any FBOs should be offered the same services and support, regardless of whether they are interested in discussing faith. She also clarified:

> Even when I mentioned our drop-in space, it's for females identified. We're not talking about only cis women or cis girls; we're talking about trans women as well. And under the broader case management, we serve anyone who's experienced trafficking. All genders. All religions. And we do not have a requirement that someone believes what we do. And there's no obligation to go to church; there's no obligation to engage in faith-based programming.

According to Amanda, many FBOs refrain from screening survivors for service eligibility before assisting them.

> Even if a rescued survivor did not meet our criteria for a human trafficking case, we wouldn't put them out on the street. We're not going to be like, "No, we're not helping you because you're not a human trafficking victim." And that's a difference between community-based organizations and governmental programs, for which sometimes you've got to answer over 50 questions before they even give a survivor a meal.

Isaac explained that faith motivates FBOs' staff to do the work and is not a condition for survivors to receive services. As he explained:

> We don't expect the people that we serve that they have to be Christians. And they do not have to ascribe to any form of Christianity or Bible studies; we're not trying to proselytize anyone. It is due to our faith that we serve, and due to our faith, we serve anyone who is in need. That does not go contrary to any funder that may be out there. But we hope funders will realize that we are not a Christian-based organization that requires people to accept our faith to receive our support.

Most respondents reported that their agencies were called to support anyone, regardless of whether they shared the same faith. There should not be any tension, conflict, or dilemma in choosing to serve and provide support to people in need. Isaac said:

> Our approach to working with anyone is to serve whoever's in the seat. And that way, we look at people, and serve them based on their humanity. We're not concerned if someone may be in the LGBTQ community; we will serve them. If the person is transgender, we will serve them however that person views themselves. However, they view their gender or their sex, or whatever. None of those things should impact our ability to serve. We don't see those as they relate to our work; we don't see those as issues or ethical dilemmas.

Likewise, Sara argued that discriminating against survivors based on faith or other values disrespects human dignity, and doing so is unethical and unhelpful for survivors. A fundamental principle of her organization was to assist people without considering faith.

Survivor Self-Determination in Service Provision

Survivors' self-determination was reported as crucial in deciding when or whether they needed faith-related services. They had to choose how and when they wanted spiritual support or guidance. Discussing how her agency upheld survivors' self-determination in their needs, Lisa explained:

> Basically, we meet people where they're at, rather than talk about where they've been, and look at how we can help these people get out of the exploitation aspect of their situation and how we might be able to support them so that they can make better choices or lead a more fulfilling life, and really just not focus on those things that could be controversial issues like that.

According to Jenna, whose agency operates with federal grants, faith topics are discussed only when survivors raise them and wish for related resources. As she explained:

> For us, the way that federal funding works, where we're limited as a faith-based organization, is that we can't initiate the conversation about faith. But if someone is seeking to connect with whatever their faith community is, we can ensure they're connected to the right resource. From a trauma-informed perspective, we want to ensure that people's needs are met and that they're ready to have conversations anyway. So, it doesn't feel like a conflict there.

Some FBOs would use their interfaith networking resources to orient survivors in their search for faith guidance. Amanda described how her FBO accommodated the faith needs of the survivors they served:

> We have a policy that whatever your faith is, we will work with it. We have relationships with the local communities of other faiths. And we sit on an interfaith panel to help people get access to the faith they mostly work with that works for them. Because while we have our faith, it is not for us to force it upon someone else. We'll help them to manage their faith. And we'll find them mentors in the community who have that faith that can be there for them.

Yet, Amanda was among those who believed that, in the long term, finding their own way into faith could be helpful for survivors in the process of rehabilitation from the exploitation they had endured. As she explained:

> If they have no faith or believe in nothing presently, they could explore different faiths and test things out because we think that part of the healing process is the ability to forgive yourself and those who harmed you along the way and to get to full healing. Otherwise, the root of bitterness takes place. And when people get bitter, they become very insular and find it difficult to move forward. So, the issue is that it doesn't matter what your faith is. Are you able to forgive? Some faiths have edicts within them that make it harder.

Survivor-Centered, Trauma-Informed Services

Six of the 14 FBOs in the study reported using survivor-centered and trauma-informed approaches in services provided to trafficking survivors. For instance, in its case management service, Isaac's FBO used a combined trauma-informed and strength-based model in its services with male survivors. Likewise, Megan's FBO had a trauma-informed mentoring program that provided survivors with a safe environment:

> Our biggest strength is our trauma-informed mentorship program. Survivors feel like they have a voice, and we're helping their voice be heard through our training, and so they're receiving a lot more healing knowing that it's helping other individuals. The mentorship program is what we thrive in.



Amanda described how her FBO helped trafficking survivors to build trust:

> Our different volunteers work with rescued victims to gain that trust. So, continuing those conversations day by day, maybe providing small things like Uber deliveries or food deliveries or electric payments or whatever, to start the sort of cycle of trust; eventually, they will choose to join the program permanently.

Diversity in Care Approaches

Three respondents indicated that a survivor-centered perspective and a clinical approach are necessary for providing services to trafficking survivors. Jean argued that using a survivor-centered or clinical approach depends on the specific situation of a survivor. She stated that:

> There are different schools of thought about how care is done. Some people come from a very survivor-centered approach where they believe that the victim makes all of his or her own decisions about the care they need. Now that's very different philosophically from a highly clinical approach. In clinical practice, you're going to have a professional saying, 'These are the issues; this is how we do care; this is what the survivor needs.' So that's other-directed versus survivor-directed care.

Further explaining her FBO's philosophy of care that blends a survivor-centered approach with a clinical one, Jean stated:

> We understand through our experience and research that at different phases of recovery, the survivor is more empowered. In the initial stages of recovery, the survivor has fewer resources, is aware of fewer options, and has a history of making bad decisions. So, there needs to be a more structured approach in the beginning. But certainly, aspirationally, we want to move in the direction of the survivor being his or her own agent.

The question is to know what a service provider would do in a situation where they had a survivor, who, at the beginning of the care they received, would like to smoke "weed" every day, and that is what they wanted. Choosing between a survivor-centered or clinical approach does not move the provider or the survivor in a healthy or helpful relationship or direction. Thus, FBOs will always face challenges in accommodating some choices made by survivors, especially those dealing with addiction. They need to develop strategies to enhance their anti-human trafficking work.

3.4.3. Atypical Funding for Anti-Human Trafficking Work

An important finding is that participant FBOs tended to eschew public funding while prioritizing private funding intentionally.

Limited Reliance on Public Funding

The findings highlight issues with access to public funding among FBOs for any of their services. Only one of the 14 FBOs reported relying primarily on public funding for its trafficking services. Another FBO had received public grants in its early years before relying exclusively on private funding. All the respondents agreed that FBOs face many obstacles in accessing public funding resources for their anti-human trafficking work. There are too many requirements and restrictions in federal funding, making it hard for FBOs that do not have a grant development team to pursue public funding. Discussing her FBO's difficulty in accessing public funding, Jean reported that they conducted a study that showed that very few FBOs have ever received federal funding for their anti-trafficking work. She had conversations with several US Congress leaders about how federal appropriations have been written, making it challenging for FBOs to apply for federal funding successfully. Describing reasons why her FBO avoided public grants, Kathy stated:

> We did prefer to operate without federal funding. It's very, very top-heavy logistical management, and also, just frankly, a lot of time invested in, get it in, in

applying for it. And the funding just wasn't that much; anti-trafficking funding is still incredibly small compared to other types of funding streams. The amount of effort to get it did not equate to the amount you get. And then, when you get it, it's a lot of work. And so, we just found that the equation didn't work for us.

Prioritizing Private Funding

All 14 FBOs, except one, primarily targeted private funding for their anti-human trafficking work. Private funding sources described include foundation money, funding from corporate businesses, funding from churches, individual donations, fund-raising, charitable donations, and member dues. Amanda's FBO relied entirely on private funding to avoid survivors the difficult experience of having to testify for investigations or prosecutions of trafficking cases:

We're entirely privately funded so we don't have to worry about forcing trafficking survivors to testify with the DOJ or anything like that. We rely on our community to fund us. Our community knows our duty of care is 100% to the survivor.

Four participant FBOs successfully secured multi-year, stable, private funding support. Describing her FBO's funding sources, Kathy said:

Our funding sources are long-time donors who feel more comfortable contributing to an anti-trafficking effort that a faith-based organization leads. Or maybe they continue to donate to S.H. because we are constant in our mission. A couple of our largest donors are not people of faith, but they know they can count on our ethics, and that's tied to our faith. So, that's where that trust factor comes in.

Some FBOs were able to support others with funding. For instance, Nicole's FBO indicated that her FBO offered small grants to partner organizations to which it referred survivors for direct services.

Yet, substantial funding for half of the participant FBOs primarily came from fundraising, charitable donations, members' dues, and community contributions. Daniel, representing the only business-based FBO, said his organization's funding came from its wealthy members' dues and foundations. As he explained,

For trafficking work primarily, first of all, over 50% of our budget is member dues. And that's based on the assets, investments, and assets under management. So, if you're a small religious organization, maybe with $2 million invested, you would pay very little. If you're a big investor, then you would pay more. But we also got supplemented by Humanity United, which has funded much work against human trafficking. And they're still funding us. There's another foundation, the Open Society Foundation, which supported us. And then there is also another called CORTICUS, affiliated with the family that owns CNA Foundation.

Reflecting on the substantial level of human trafficking-related services FBOs provided, Jean argued that FBOs could be considered any state or nation's cheapest partner for anti-trafficking policy implementation.

The findings show that Catholic FBOs were arguably financially stable and did not often depend on funding from other sources. Describing the unique case of her FBO's funding stability, Hannah explained:

Sisters are just very financially well off. They have had many people who give them donations. Donors trusted the sisters to allocate the money where they felt it was necessary. Thus, sisters allocated substantial resources to anti-trafficking work. Money was never an issue for us. I know it is for so many, 99%, but it was never an issue.

Stacey's FBO also relied on a stable funding source from a religious order of sisters in the Catholic Church.

Yet, despite challenges in accessing public funding for sustainable funding, some FBOs contemplated diversifying their funding sources by applying for grants at the city, county, state, and federal levels. As Isaac stated, "We are definitely looking to diversify that and begin seeking more local, state, and federal funding opportunities. We recently received a grant from the City of Chicago to provide rapid rehousing services." One FBO conducted income-generating activities, with the participation of survivors, to increase its financial resources for anti-human trafficking work. As Megan stated:

> We actually have shops as well. So, what we do is we have items made by survivors. We roast our coffee, make clothing, and design apparel. Then, we partner with other anti-trafficking agencies with items made by survivors. And we sell those items in different stores throughout our region. We do pop-up shops, and we actually have the survivors help us too, helping with making the coffee, helping with all of our tagging and labeling, and taking things to the shops. And that gives them job skill training, as well as being able to put something on their resume, and the net profits all go back into our organization.

### 3.4.4. Salient Challenges in Anti-Human Trafficking Work

The findings highlight major challenges participant FBOs faced in their anti-human trafficking work. While nine respondents reported financial difficulties their agencies encountered, all 16 participants raised other issues that impact their FBOs' anti-human trafficking work, including gaps in victim services and resources, misperceptions about FBOs' engagement in anti-human trafficking work, ethical issues around converting survivors to religion, and faith as an enabling factor for exploitation.

Financial Challenges in Anti-Human Trafficking Work

Respondents pointed out that their FBOs sometimes faced various financial challenges in providing human trafficking-related services, including funding instability and limitations impacting service quality, costly service provision, competition for funding, and faith's influence on access to funding.

Funding instability

FBOs are sometimes concerned about running out of funds for the work while survivors still need consistent care and more qualified staff for trauma-informed services. Jenna explained that:

> Funding for any anti-trafficking work is hard for programming in general and to have consistent long-term services. If you are talking about relying on federal funding, those funding cycles are about every three years. So, you get two years in, and you're already thinking about what grants you can apply for. We have really incredible staff who are super motivated and do great work, but we don't want them to get burned if we don't get refunded. So, funding is always a challenge.

An issue that deepened funding for anti-human trafficking work was that organizations doing the same work in the same city or state would compete for funding from the same funders instead of collaborating for services.

To address funding instability and sustain their services, respondents reported that their FBOs relied primarily on volunteers, as mentioned above. As Amanda explained, "Our funding goes up and down all the time, but we are heavily volunteer-run in all our places. We're lucky to have a team of amazing volunteers who are the backbone of our organization." Relying primarily on volunteers helps FBOs to offset limited financial resources for anti-human trafficking work. Sara, from the only Jewish FBO in the study, said:

> We're volunteers; we don't have that budget. Very little budget. Our membership in Chicago North Shore is about 650/700 active volunteers. There are about eighteen of us who are volunteers on board. So, you have a couple of hundred enthusiastic volunteers in and out at any time.

Faith's influence on access to funding

Sometimes, FBOs are reluctant to accept funding with requirements not aligned with their faith. Jean said her FBO would not accept any funding support from businesses such as Amazon because it was claimed that Amazon allows companies that thrive on the sexual exploitation of children by using its online platform. Other FBOs would not accept a grant from any entity that intended to put too many restrictions on how to use the funding to assist survivors. For instance, Amanda's FBO would accept a grant only if its conditions did not hamper how the agency implemented its victim services or delivered its curriculums.

However, funders would discriminate against FBOs of specific faith denominations in awarding grants for anti-human trafficking work. For example, according to Sara, for unsubstantiated reasons, some grant makers presume that organizations of the Jewish faith denomination do not need financial support for their work with vulnerable groups. This misperception negatively affects the capability of her FBO to raise funding for its anti-trafficking services.

> I had numerous funders who would come back to me and say, 'The Jewish community doesn't need more money. You're a Jewish organization.' I worked for Jewish Child and Family Services, and they said, 'You're a Jewish agency. You have plenty of money.'

Gaps in Victim Services and Resources

Respondents identified several challenges in victim services and resources, including the following: addressing other needs associated with trafficking; difficulty providing mental health services for survivors; difficulty in sustaining trauma-informed services; gaps in services for male survivors; and difficulty in providing long-term housing for survivors. Since no agency could have enough resources to provide comprehensive services addressing victims' multiple needs, FBOs often had to determine areas of service they could focus on with their limited resources. Access to mental health services was found to be problematic not only because of the limited availability but also because of lengthy delays for mental health services. Amanda explained:

> Our biggest challenges are access to good mental health care at the scale we need. That's one of the most challenging things we face. Unless the person is acutely suicidal, it can take six to eight weeks to get into the rotation for care from a psychiatrist. Sometimes that wait is too long. But the vast majority of the population we work with is uninsured or underinsured. So, it is much more difficult.

While shelter services are available for women and girls, there is a significant need for safe houses for trafficking survivors who need long-term services. Amanda explained:

> The second one is housing. Not just accommodation, like it's very easy to put people into independent care accommodation where we rent an apartment, or we put them up in a hotel or whatever, but what is difficult is when they are not ready for independence, and that 24/7 care has to happen. That type of housing is much harder to fund, to keep going, and all of those things. That's probably the other thing that's so very difficult.

Though many of the FBOs in the study valued survivors' self-determination and participation in interventions, Jenna, one respondent, observed that it was challenging to involve survivors in programs addressing their needs because lived experience is not comparable with professional skills. As she said:

> One of our biggest challenges is the delicate path we have to walk in working with survivors and allies. We have a lot of survivor-led organizations that we try to come alongside and help move forward. But, very frankly, while survivors have life experience, sometimes they don't have sufficient organizational experience. We need to be able to support the work they're doing and the whole

relationship thing to try and be colleagues, and advisers without stepping on toes or overstepping our bounds.

Misperceptions about FBOs' Engagement in Anti-Human Trafficking Work

The findings highlight distrust in interactions with FBOs, suspicion about FBOs' motives in anti-human trafficking work, and the questioning of FBOs' legitimacy for anti-human trafficking work. Respondents concurred that people outside the faith communities think that FBOs have a hidden agenda in doing anti-human trafficking work. As Lisa explained:

Whenever you're a religious organization, people are worried that you have a religious agenda, that you're either going to force conversion on people, or you're going to make it conditional for purposes of receiving help. And we had to make it clear that we have no agenda like that. We're just living out our mission. We're not looking to convert people. So those are probably the credibility and the fear of conversion.

Sometimes, FBOs were not taken seriously despite the frontline efforts in addressing trafficking. Besides serving trafficking survivors, respondents reported that their FBOs had to defend their legitimacy and credibility in doing anti-human trafficking work. Describing the early experiences of her organization in the field, Lisa further argued that:

The first challenge in doing human trafficking work is credibility. Initially, we were called the church ladies. That's how we were identified. It's like, 'Oh, here's the church ladies now.' Well, that doesn't give us much credibility. That doesn't say much about what we're doing, but we went along with it because we thought it was fine. However they identified us, it was a start. But then they came to realize what we can do. And so, they don't call us the church ladies anymore, except in a fond memory kind of way, like, 'Remember when we used to call you the church ladies'.

Participants reported that many people working in FBOs found it frustrating that they often had to justify their motivation in taking on anti-human trafficking work. Jenna explained FBOs' tenacity to continue anti-human trafficking work the best way they could while showing openness for partnership with other stakeholders.

Sometimes people make assumptions that we would require certain participation of survivors in whatever faith-based programming. I think it takes a long time to build relationships with community partners so that they can see what our intentions are, and what our program is. It takes time. We don't want to tell people, 'Just trust me.' If we expect to build rapport and trust with partners and survivors, we should also be willing and able to do that. So, I think that's an important kind of barrier that we're always working to overcome.

Amanda further elaborated on the recurrent challenge for FBOs to provide reasons at every step of their engagement in the anti-human trafficking movement:

Increasing the legitimacy of FBOs in the anti-trafficking movement with policy-makers and funders is the biggest challenge. At the end of the day, we've got some pretty crazy faith-based organizations out there doing some pretty crazy stuff. I would know this because I have met them, and they make it difficult for the others striving to do it right. And there is much distrust. There is presently a lot of distrust out there between the federal and state organizations and the FBOs. The only way to combat that is for the state and federal organizations to actually see and experience what's going on in the FBOs.

Ethical Issues around Converting Survivors to a Particular Faith

All 16 respondents concurred that it is counterproductive that any FBOs providing anti-human trafficking services would push their faith onto trafficking survivors. After

experiencing a traumatized situation of captivity, survivors may not be ready or willing to comply with a whole new set of faith requirements, which can have a re-traumatizing effect and result in them distrusting those who were supposed to help them.

> I would be cautious about when assistance crosses into the perception of 'I'm trying to save you because you need to get out of this horrific situation you're in,' or 'I want to save you, and then there's just a certain way you need to be in this religious structure.' So, I think that when religious organizations veer into extremism, that is unhelpful. (Amanda)

Thus, preaching to trafficking survivors could hamper their self-determination and trust-building. As Annie observed:

Some FBOs tend to proselytize because "they want to save and rescue people. And in many faith traditions, there is this belief that the only way to save someone is to bring them to a particular faith tradition," Jenna said. According to Leah, another participant, it is not only unethical for organizations in anti-trafficking work to use faith as a bargaining tool but also harmful to survivors and the faith tradition.

Three respondents reported knowing FBO leaders, trafficking survivors themselves, who would unintentionally project their past trauma into their work as they tried to convert survivors. Likewise, for being in constant contact with trafficking survivors and intensely listening to their traumatic captivity stories for too long, respondents said that the staff in some FBOs experienced vicarious trauma, which is a process of change resulting from empathetic, prolonged engagement with trauma survivors. As one respondent explained, "What we've seen is a lot of times people at anti-trafficking organizations usually have had their own trauma and abuse. And they're bringing their own stories into the way they lead." Overall, the findings show that not having clear service standards and boundaries for separating assistance services and faith could harm any therapeutic relationship with survivors who do not want to be in another situation that they have no control over.

Five respondents acknowledged that religious leaders have sometimes used faith to coerce people into an exploitative situation. People may be so vulnerable and traumatized that they will accept any conditions or do anything to obtain shelter, food, and water, resulting in situations where religious leaders can take advantage of and revictimize them.

> We have traffickers who use the Bible to exploit other people. We have traffickers who have been convicted, who were leaders in their own church. So absolutely, I think anything can be manipulated, misused, and abused. That isn't what our faith actually is. That's not a true biblical teaching, but absolutely anything can be distorted. (Amanda)

### 3.5. Improving FBOs' Input in the Fight against Human Trafficking

Findings from respondents' perceptions on ways to improve FBOs' contributions to the fight against human trafficking are summarized in three subthemes: need to expand FBOs' presence in the field; importance of addressing misperceptions about FBOs; and need to evaluate FBOs' anti-human trafficking programs.

### 3.5.1. Importance of Expanding FBOs' Presence in Anti-Human Trafficking Work

Respondents discussed a few ways to expand FBOs' input in the fight against human trafficking, including: giving due attention to FBOs' anti-human trafficking work; need for collaboration between the public, stakeholders and FBOs; reaching out to more FBOs for input in anti-human trafficking work; importance of survivors' self-determination; and accounting for FBOs' distinctive competencies in policy implementation.

### Importance of Giving Attention to FBOs' Input in Anti-Human Trafficking Work

Respondents perceived that, despite their active contributions to the fight against human trafficking, FBOs face issues of legitimacy in the field. Using the example of Texas

state's outreach efforts to FBOs, Jean argued that the extent of FBOs' contributions depends on the local, state, or federal political environment. As she explained:

> If you take the state of Texas which has the second largest number of shelters in the US, they have a big trafficking problem and it's getting worse, but their government is very intent on engaging the faith-based sector. So they have always invited us to the table. So, we're gonna see better outcomes in a state like that.

This respondent argued that an adversarial relationship between the policymakers and the faith-based sector sometimes portends serious issues in service responses to human trafficking. Connecting FBOs to public and criminal justice stakeholders is important, because, as Annie argued, "It could be helpful to connect FBOs with governmental bodies, with law enforcement, and with any sort of legal body that is working on human trafficking." Respondents also believed other stakeholders in the field of anti-human trafficking work could learn from FBOs' approaches to advocating for policy change and providing services. As Leah argued:

> Maybe we could be used as a model, if they are seeing success with what we are doing, not to recreate the wheel, we have so many programs up and running. We'd be glad to share them. And not all of them are oriented towards faith. We access the guidance of researchers in statistics, literature, and science, and what's happening in our community.

Respondents perceived that FBOs have become a powerful catalyst to reckon with in the anti-human trafficking movement. As Stacey explained, "They have to recognize that we're key players. I think faith-based organizations have to be comfortable that they do have power. I think lawmakers need to see that we have numbers and can make a difference".

Increasing Survivors' Self-Determination

Respondents stressed not only the importance of survivors' self-determination in the assistance services they could access but also the importance of survivor-informed services. Emphasizing the importance of the survivor's voice in FBOs' work, Stacey pointed out that:

> In any work around anti-trafficking, the importance of bringing the survivor to the table cannot be stressed enough. It's a very difficult thing to do because you don't want to exploit them again, yet their lived experience is crucial to finding solutions and improvements.

Relating to his over-four-decade's worth of experience of advocating for anti-trafficking policies within the corporate world, Daniel said:

> Our work about human trafficking needs to be informed by survivors, not by what we think. Thus, the strategy we are recommending companies is not to try to do this all but to listen to those with the life experience, the survivors; don't just listen to their stories. They're their strategists. Therefore, we should ensure that they help shape the movement going forward.

3.5.2. Filling the Gaps in Research on FBOs' Anti-Trafficking Programs

Subthemes regarding research concerns include the need to evaluate FBOs' work on human trafficking; the need for more data than stories to reinforce the legitimacy of FBOs; and the importance of understanding the similarities and differences among FBOs.

The lack of evaluation of FBOs' anti-human trafficking work was pointed out by participants as a major weakness that hampers the value and scope of their distinctive contributions. According to Megan, there were not enough FBOs doing research because they were so busy on the front lines, "working with the people in the trenches, rolling up their sleeves, and doing the work. And there are not enough people that are competent to do the research and do the investigations and things like that." She argued that research about FBOs was often conducted by non-faith-based organizations or individuals who

have little understanding of FBOs'work. Likewise, Jean, whose FBO was one of the rare ones to have conducted research on human trafficking, observed that the lack of research about FBOs' anti-human trafficking work negatively affected their credibility with other groups of stakeholders. She explained that:

> There is more need for research to improve the legitimacy of FBOs in the anti-trafficking movement. We have to show the efficacy and the impact of these agencies across the board, whether they are faith-based or secular; we need to be able to measure because then we cannot determine whether one model exceeds another. We also need to behave with more data-driven decision-making. We sometimes operationally do so much on feelings or will, or these fewer concrete measures. I have always been told that the faith-based community relies too much on stories and not enough on statistics.

Jean argued that FBOs' anti-trafficking program evaluations could help improve the services provided to survivors. Research could help the faith-based sector to communicate outcomes, measure them, track them, to be able to tell stories and spreadsheets, to be able to say:

> Well, if seventy-eight percent of our survivors are coming in with addiction issues, what are we doing to lower that percentage.' We can still tell a story, but we need data. I think research studies like yours [that is, the current study] are very important. So, that's raising understanding and awareness.

Like Jean, Megan claimed the current study might allow other stakeholders to gradually understand and show interest in FBOs for their distinctive competencies in contributing to the fight against human trafficking. As she stated, "I love that you are researching and learning what faith-based organizations are doing and the why behind it. And the more people are aware, the more calls we know that we'll receive, and we want to be open, ready, and available to meet those needs." It is also important to understand the similarities and differences among FBOs in the field and evaluate their input in anti-human trafficking policy implementation because, as Isaac explained:

> All faith-based organizations are not 100% alike. Therefore, there will take some effort to get an opportunity to learn and understand the various faith-based organizations and not just lump them all into one homogeneous group; there's much diversity.

### 3.5.3. Addressing Misperceptions about FBOs' Legitimacy and Capacity

Suggestions for addressing misperceptions about FBOs' legitimacy and capacity for anti-human trafficking work are highlighted through five subthemes, including: recognizing FBOs' legitimacy, developing and sustaining partnerships, helping more FBOs to receive training on human trafficking, assessing FBOs' competencies FBOs can contribute;, and addressing difficulties in developing interfaith collaboration. Two of these subthemes are described below.

### Reckoning with FBOs' Legitimacy in the Fight against Human Trafficking

Four respondents shared that FBOs do not need to prove themselves as legitimate stakeholders in the fight against trafficking. Illustrating this point with her organization's leadership in policy advocacy, Nicole said: "We're considered hugely legitimate already. For over ten years, we've issued report cards to every state on how they're on the impact of their laws against trafficking, and now their policies, and that's like a standard for the entire country".

Using his extensive experience as a leader of an FBO consisting of Christian investors and shareholders, Daniel suggested three strategies to advocate against human trafficking in big corporations. First, assessing the content and impacts of companies' policies on vulnerable people is important. As he said, "It was not just to meet with companies and say, 'Well, good work;' we do look at their labor policies and practices, how they impact people,

what their impacts show, and if what they're doing is abusive." Second, the leverage of faith-connected investors and stakeholders is critical in holding companies accountable. Finally, knowing about the culture of target companies before advocating for change within it is important.

## 4. Discussion

### 4.1. Faith's Influence on FBOs' Anti-Human Trafficking Work

Three important themes require discussion: faith's importance in the anti-trafficking work, Christian organizations' prominence in anti-trafficking work, and uniqueness of having anti-trafficking FBOs in the for-profit system.

All 16 respondents cited sacred scriptures to describe the beliefs and values guiding their FBOs in investing time and resources in this work. This finding corroborates the literature indicating that faith provides FBOs with a strong call for anti-human trafficking work [1,10,11]. Pureskal [11] found that Christian anti-trafficking organizations within and outside the USA identified the Bible as the inspirational force that guides their engagement. Central to most faiths is the belief in the innate dignity of all humans. Likewise, Barrows [10] showed how sacred scriptures are used to justify the engagement of major FBOs involved in the US-based anti-trafficking movement, discussing the Torah's influence on Jewish FBOs' anti-trafficking drive, the New Testament's validation of Christian FBOs' commitment, the Qur'an's guidance for Muslim organizations' motivation, and the Bahá'u'lláh's inspiration in Bahá'í's activism. T'ruah [13] argued that the anti-human trafficking work of Jewish FBOS is historically connected with the Jewish people's historic journey to freedom from slavery.

The overwhelming representation of FBOs of Christian denominations among participant organizations (13 of 14) provided an opportunity to understand the substantial influence of the Bible's teachings on the passion and mission of many organizations fighting human trafficking. Many NGOs that have influenced and shaped the US anti-human trafficking policy were of Christian denominations [1]. Even outside the US, Christian organizations are the most prominent ones in the fight against human trafficking [6,8,11,14,21]. The moral commitment of many Christian FBOs in the fight against human trafficking draws, in significant ways, on the precedent set by abolitionists who worked in Britain and the US during the 18th and 19th centuries to end chattel slavery, a race-based institution that permitted the legal ownership of one human being by another [19]. According to Zimmerman [19], Christian FBOs are major partners in the anti-trafficking movement because of the moral clarity that they bring to the issue. Christian FBOs' anti-trafficking activism and advocacy are animated by their overarching goal of reinstating traditional gender roles while crusading against the sex trade in all forms {Zimmermann [19].

Another reason that could justify the frontline role of Christian FBOs in their anti-trafficking work in the US is a traditional over-representation of Christianity in the nation. Seven in ten Americans (70%) identify as Christian. Nearly one in four Americans (23%) are religiously unaffiliated, and 5% identify with non-Christian religions, including 1% Jewish, 1% Muslim, 1% Buddhist, 0.5% Hindu, and 1% who identify with other faiths [36]. The finding of three participants who had quit lucrative businesses or high-ranking lawmaking positions to establish FBOs of Christian denomination indicates a strong influence of faith in the passion of some Christian FBO leaders to address human trafficking. The three respondents reported that people outside faith communities found it challenging to understand FBOs' enthusiasm for taking on anti-human trafficking work, which is often perceived as having a hidden agenda.

The data from only one FBO of a Jewish denomination in this study provided an opportunity to partly understand the motivations and perceptions that FBOs of other faith denominations shared with Christian ones. This FBO was reported as part of the National Council of Jewish Women, which rescued and assisted immigrant girls and women trafficked or at risk of being trafficked on Ellis Island (New York) in the early 1900s. For Jewish FBOs, the fight against human trafficking reverberates Jewish people's historic journey to freedom from slavery [13]. Yet, in contrast with Christian FBOs, the anti-trafficking work

of Jewish FBOs has not been well explored in the literature. Overall, data from FBOs from denominations other than Christian and Jewish ones could help understand and contrast the motivations and engagement of different groups of organizations in the anti-trafficking movement based on their respective sacred scriptures and histories.

Before the present study, no literature existed on anti-trafficking FBOs operating in business. The participation of one FBO made of business shareholders is noteworthy. It shows how faith could be successfully combined with business as leverage to advocate for and demand policy reforms addressing human trafficking inside major corporations and companies. Advocating for anti-trafficking reforms in corporations could be helpful when it is conducted from inside the business environment. Members of this unique FBO could provide it with the credibility and insider knowledge necessary to expose companies' failure to address exploitative situations in their businesses and supply chains and persuade them to reform to address human trafficking.

### 4.2. FBOs' Capabilities for Anti-Human Trafficking Work

A few themes from the findings on respondents' perceptions of the competencies of FBOs for anti-human trafficking work are important to highlight. The findings indicate that traditional FBO networks within communities and congregations allow them to tap into diverse competencies and assets necessary for prevention-focused services such as awareness-raising and outreach campaigns. According to Harrelson [6], FBOs make invaluable, unique contributions to human trafficking because they have access to extensive human (e.g., volunteers) and financial resources (e.g., private funding). They enjoy legitimacy in communities and are often headed by moral leaders perceived to be trustworthy [6,30]. FBOs' influential role in the passage of various anti-human trafficking policies in the UK and the US could be partly explained by their ability to mobilize groups and communities around social problems [2,10,28,30]. Respondents reported that FBOs' ability for outreach and awareness-raising on human trafficking was often valued by law enforcement. The finding on law enforcement agencies reaching out to FBOs for help indicates a high level of trust in the latter's capacity to mobilize substantial community support. Four FBOs in this study often assisted the police and the judicial system at various levels by being go-betweens with survivors or providing food, shelter, and other assistance to those who needed to testify in prosecutions.

A few protection-focused capabilities highlighted in the findings are notable. The FBOs in the study reported having competencies for survivor-led and survivor-centered trafficking services. The results suggest that FBOs have traditionally valued survivors' input in programs for a long time. As the findings show, trafficking survivors, who graduated from an FBO's assistance program, preferred to stay in the organization and help either as staff or volunteers. Some survivors are the FBO founders and/or leaders who participated in the study. Having survivors as key staff in an anti-trafficking program is important because it allows input from lived experience experts who may better understand survivors' service needs [37]. They know firsthand what is needed to improve anti-trafficking responses, and their input is critical to ensuring anti-trafficking policies reflect perspectives that only those with lived experience can provide [37]. Their presence on agency staff could also help other survivors develop trust in service providers. Only during the last ten years have US departments and grantmaking agencies realized the importance of having survivors in the management team of anti-trafficking programs. In recent years, they have aggressively advocated for any entity, whether a government, business, or civil society organization, to seek meaningful input from a diverse community of survivors at each stage of a program or project [37]. Consequently, more and more grant makers require the inclusion of trafficking survivors in implementing research projects or service programs that receive public funding.

It was reported that FBOs could provide long-term services to survivors because of two assets: expertise in shelter and safe housing for survivors and reliance on private funding. Housing accommodation could be a vital part of long-term services for trafficking survivors.

Respondents claimed that FBOs owned approximately every seven out of ten shelters and over half of the safe houses for female trafficking survivors. This claim was corroborated by a survey report showing that over 80% of the shelters available for victims of sex trafficking in the United States are owned by FBOs [38]. FBO's competency in housing should be further examined and accounted for in anti-human trafficking policy implementation.

Participants perceived FBOs' tendency of prioritizing private funding as a major advantage in providing long-term services to survivors because it allows flexibility in services and extended service time. FBOs' tendency to rely on private resources led some respondents to argue that FBOs make significant contributions to human trafficking policy implementation, even though this input is hardly accounted for in policy implementation evaluations. As Jean observed, "Faith-based agencies are a state or a nation's cheapest option. That's just a practical reality because somebody else is paying for all of that care. FBOs in this work provide a valuable option for any state." However, Lewis et al. [8] made a pertinent point that by relying primarily on private funding to fill the gaps in services needed by trafficking survivors, FBOs might be unconsciously encouraging cuts in public funding for anti-human trafficking policy implementation.

*4.3. Highlights of FBOs' Experiences and Contributions*

4.3.1. Contributing to Prevention, Protection, and Prosecution Efforts

While FBOs are known for their awareness-raising and outreach services, three prevention service areas where they may be less expected to intervene are training, ICT, and policy advocacy, which most FBOs in this study reported. Human trafficking-related training was not only the most frequent activity reported by 10 of the 14 FBOs, but also their training activities targeted diverse groups of stakeholders who have a crucial role to play in the prevention of human trafficking, including health agencies, schools, religious congregations, businesses, and ethnic associations. Training activities were also conducted to increase the competencies of FBOs' staff and volunteers. FBOs' experience using advanced online strategies to prevent human trafficking and protect potential victims could be a key finding. Since a substantial amount of human trafficking activity has moved online, ICT is increasingly utilized to find and reach out to potential victims. Notably, it was reported that three participant FBOs were using ICT tools to monitor sex trafficking ads online and track transactions for potential victims on the Dark Web. One FBO was experienced at screening, analyzing online data on sex trade ads, and running algorithms to identify geographic locations where sex trafficking was taking place. ICT played a crucial part in another participant FBO's efforts to reach out to and assist male survivors of trafficking.

Another significant contribution made by participant FBOs was in housing for trafficking survivors. While one participant FBO was reported as one of the big agencies providing shelter services to trafficking survivors nationwide, and two others were reported as being among the few providing long-term transitional safe housing services for female survivors. The distinctive aspect of another FBO was that it uniquely specialized in preserving family units by keeping women survivors with their children and their pets, if applicable. Respondents reported that survivors' self-determination in reaching out for services such as safe housing contributes to collaborating with staff for rehabilitation, resulting in high retention and graduation rates from FBO-owned housing programs. Housing services for trafficking survivors provide a glimpse of the leading experience of FBOs in this area.

Beyond direct services such as housing, participant FBOs' input in policy reforms at national and international levels could be considered substantial. According to Gee and Smith [1] and Barrows [10], FBOs have historically influenced the development of anti-human trafficking policies in the UK and the US. Weitzer [20] argued that FBOs, especially those of Christian denominations, would use sex trafficking to push their anti-sex policy agendas. In the present study, one FBO has had a frontline role in policies addressing domestic minor sex trafficking (DMST) during the last 10 years. Through national conferences, interfaith committees, and policy development and implementation monitoring at the federal and state levels, this FBO has successfully influenced the development of new

policies such as Safe Harbor legislation in many states and monitored the implementation in all 50 US states. Beyond the successful policy reforms that it contributed to in three US states, a second policy advocacy FBO was successful in advocating for anti-trafficking policy reforms in for-profit companies at the international level.

Human trafficking investigations and prosecutions could be a major aspect of anti-human trafficking work that NGOs, particularly FBOs, would be least expected to contribute to. Still, law enforcement, judges, and prosecutors would contact three participant FBOs in this study for help. Two of these FBOs assisted major criminal justice stakeholders, such as the Department of Homeland Security and the Attorney General's Office, etc., in the process of court cases in which trafficking victims were key witnesses. Barrows [10] identified only one FBO experienced in assisting law enforcement in investigating human trafficking cases (i.e., Hope for Justice USA). This FBO, which could not participate in this study, has often assisted regional law enforcement agencies in investigating many human trafficking cases [10]. Criminal justice system stakeholders tend to reach out to FBOs, possibly because of the latter's experience in mobilizing community resources, their ability to show compassion and gain survivors' trust, and the moral influence of religious leaders in communities.

### 4.3.2. Applying Ethical Values in Services

The perception that many FBOs utilize their services as a means to convert trafficking victims to religion is reported in the literature [11,20,22,24,25]. Yet, all the participants in this study counterargued that efforts to convert trafficking survivors to a specific religion are counterproductive because such actions can hamper survivors' self-determination and trust-building, and they can become re-traumatized. They also perceived situations where religious leaders use their power to exploit vulnerable individuals as unethical. Such professional misbehavior negatively impacts the work of "well-intentioned FBOs assisting survivors," as one respondent said. Before the current study, Lewis et al. [8] found no evidence supporting the pervasive perception that many FBOs would convert trafficking survivors to their faiths. However, more research is necessary to better understand the use of religion in services provided to trafficking survivors.

While respondents acknowledged that faith support could be vital to survivors' spiritual needs, recovery, and restoration, as corroborated in the literature [8,30,39,40], they also indicated that their FBOs prioritized survivor self-determination as a key principle governing their services. Survivors' self-determination helps avoid a situation of power and control of service providers over survivors. A shared view among the respondents was that survivors should decide when they want spiritual support, and faith questions should be discussed only if survivors raised them and wished for such resources. Overall, the finding on the consideration of survivors' self-determination in service provision could be perceived as part of efforts by participant FBOs to limit unwanted conversion of survivors to any faith without the latter's consent and to address misperceptions faced in taking on anti-human trafficking work.

### 4.4. Addressing Challenges for Anti-Human Trafficking Work

This study's findings highlight some crucial contributions FBOs make to anti-human trafficking policy implementation while relying primarily on private funding. Thirteen of the 14 FBOs reported rarely applying for or receiving public grants for their anti-human trafficking services because of the various challenges they faced. Only one of the 14 participant FBOs reported relying only on federal funding for its work. Four FBOs, which had leading national roles for specific human trafficking services such as policy advocacy, shelter/safe housing, and direct services, reported not targeting federal or state funding for their work due to the reasons highlighted above. All respondents reported too many hurdles for FBOs to secure public grants, arguing that public grants have too many restrictions (i.e., time limits, invasive monitoring processes, funding inconsistency and conditions, and too much paperwork, etc.), which can affect the quality of services for trafficking survivors. As the

findings show, the odds for FBOs to successfully apply for public funding and obtain it are further affected by concerns related to faith considerations. Thus, they tend to target private funding for their work because it is "more accessible, more flexible, and easier to manage and report," as one respondent explained. Operating outside public funding allows FBOs to develop services that are not resource-constrained, time-limited, and policy-restricted. Respondents perceived that it could be challenging to estimate the time necessary for recovery for survivors who have experienced trauma during their trafficking situations and to assess the cost of long-term care they may need ahead of time. They perceived that these aspects of service to survivors are rarely factored into the requirements of public funding.

However, while most respondents conveyed consistency and accessibility in private funding for their agencies' work, a few reported challenges in both funding and other resources necessary to serve trafficking survivors. Access to specialized services requested through referrals is often tricky, which puts too much stress on FBOs trying to find appropriate help for survivors, especially for health and mental health services. Moreover, the finding about respondents reporting a lack of trauma-informed services as a critical challenge contradicts an initial finding in this study showing that most FBOs used a trauma-informed care approach.

### 4.5. Perspectives on Improving FBOs' Engagement

Respondents perceived that enhancing FBOs' contributions to anti-human trafficking policy implementation calls for addressing entrenched misperceptions about their motivations and service approaches and assessing the quality and impact of their anti-human trafficking programs. Their perception that FBOs still have to fight for legitimacy as major stakeholders in the fight against human trafficking is notable. They shared the view that secular and public stakeholders misunderstand FBOs' motivations for investing time and resources in addressing human trafficking. They lack knowledge on FBOs' distinctive approaches and competencies for anti-human trafficking work. Some participants suggested that political support could be crucial for validating FBOs' input in anti-human trafficking policy implementation, as was the case in Texas, where the current government values the active input of FBOs in the state's policy implementation, according to one respondent. Accounting for contributions made by FBOs through anti-human trafficking-centered training, policy advocacy, housing services, and awareness-raising campaigns could help understand their important role in the overall policy responses to human trafficking.

Participants' call for research on FBOs' anti-human trafficking service experience and impact could be considered FBOs' disposition to have their distinctive input validated and enhanced through interagency collaboration with secular and public stakeholders. So far, a large part of the research on the anti-human trafficking work conducted by FBOs in the US is based mainly on surveys of information available online. Evaluating FBOs' human trafficking programs could help address misperceptions they face and measure the effectiveness of their service approaches while allowing for comparison with the performances of other stakeholders in the field of human trafficking.

## 5. Implications and Limitations

### 5.1. Implications for Practice, Policy, and Research

This study has important implications for stakeholders, including FBOs, secular NGOs, criminal justice system leaders, and public administration leaders. Secular NGOs could learn from the decades of experiences and contributions of FBOs to the field of human trafficking, possibly learning from FBOs' distinctive competencies and approaches to advocating for policy change and providing various types of services for the survivors of trafficking. It is essential to assess and understand some of FBOs' approaches to addressing human trafficking. The findings highlight using a system such as Maslow's Hierarchy of Needs in assisting trafficking survivors, valuing survivors' self-determination, and input in managing human trafficking services.

It is important for FBOs to develop interfaith ethical standards on trafficking survivors' freedom of belief, religion, and thought to be used by FBOs engaged in anti-human trafficking work. More specifically, state governments could take inspiration from the UK's *Slavery and Trafficking Survivor Care Standards* developed in 2014 and updated in 2018 by the Human Trafficking Foundation [40]. In October 2017, the British government stated that the *Slavery and Trafficking Survivor Care Standards* would be adopted in any future trafficking victim care contracts with both FBOs and secular organizations working with trafficking survivors. Analogous care standards should be developed and implemented in other countries, particularly the United States. Most participant FBOs in the present study claimed their agencies had policies addressing proselytizing and discrimination in services. However, having FBOs formally commit to applying evidence-based survivor care standards could help address the misperceptions they face in their anti-human trafficking work.

There is a need for more outreach to other FBOs still not involved in the anti-human trafficking movement. Strategies should be developed to harness FBOs' collective capability for policy advocacy and service provision, which could result in more FBOs' input in task forces, coalitions, and decision-making arenas. Finally, combining their strengths and resources is crucial to strengthen the effectiveness of FBOs' input in the human trafficking field. FBOs' contributions to anti-human trafficking policy implementation have been overlooked for a long time because of the lack of evaluation of their varied services' impact on the prevention of human trafficking, protection of survivors, f and even prosecution of human trafficking.

On 14 February 2021, President Biden signed an Executive Order re-establishing the White House Office of Faith-Based and Neighborhood Partnerships (OFBNP) to promote partnerships with faith-based and secular organizations to better serve people in need [41]. Recently, in collaboration with the President's Interagency Task Force, Senior Policy Operating Group on Public Awareness and Outreach on Trafficking in Persons, and Blue Campaign, the OFBNP invited more than 3500 faith-based and community organizations to get involved in combating human trafficking by adopting practices highlighted in the Department of Homeland Security's *Blue Campaign faith-based and community toolkit* [42]. In line with the purpose of the OFBNP, seven federal departments established their Centers for Faith-Based and Neighborhood Partnerships. These centers aim to foster partnerships between the government and faith-based organizations (FBOs) to increase the nation's resilience by creating trust and developing relationships. Building on the information above, the OFBNP and the US Centers for Faith-Based and Neighborhood Partnerships should consider assessing how to develop a better partnership with FBOs for strategies to address human trafficking. Despite the traditional separation between state and religion, FBOs' influence on human trafficking policymaking cannot be overlooked. There is a need for policymakers at the federal and state levels to reach out for opportunities to enhance their involvement in public anti-trafficking programs.

FBOs' input in anti-human trafficking policy implementation and challenges to their access to public funding need to be further explored and expanded. The lack of research on FBOs' anti-human trafficking work negatively affects their ability to work with other stakeholders who still misunderstand FBOs' competencies and motivations against human trafficking. There is a need for more data than stories on services that FBOs provide regarding human trafficking to strengthen the legitimacy of their work. It is important to understand the similarities and differences among FBOs engaged in anti-human trafficking work. It is also crucial to conduct comparative research on the anti-human trafficking work of FBOs, secular NGOs, and other community organizations to improve the contributions of each of these groups in implementing human trafficking policies. It would be important to further explore the new research topic of advocacy of faith-based business stakeholders against human trafficking in major for-profit companies.

*5.2. Limitations of the Study*

The study findings cannot be generalized to other populations of FBOs with experience of anti-human trafficking work in the United States because a small purposive sample and qualitative method were utilized. The over-representation of FBOs of Christian denominations among the respondents is another limitation. Yet, as the findings show, the input of a single FBO of the Jewish denomination was substantial. Information from FBOs of other denominations, such as Muslim, Hindu, and Bahaí, etc., could have improved the study's results. Given that the interviews were conducted via Zoom, the effects of online technology might have influenced the quality of the collected data in ways not fully known. Social desirability bias likely shaped some responses of participants because they were all representatives of organizations that might want their anti-human trafficking work to be perceived in a positive light. Likewise, it is difficult to know to what extent the researcher's expertise in human trafficking impacted the interpretation of the data. However, the use of member-checking and peer-debriefing strategies helped limit the potential threats to the credibility and confirmability of the findings.

**Funding:** This research was funded by the Jane Addams Center for Social Policy and Research, University of Illinois Chicago, with the Summer 2022 Research Grant.

**Institutional Review Board Statement:** The study was conducted in accordance with the Declaration of Helsinki and approved by the Institutional Review Board of the University of Illinois Chicago (Protocol # STUDY2022-0726).

**Informed Consent Statement:** Informed consent was obtained from all subjects involved in the study.

**Data Availability Statement:** The qualitative data for this study are available on demand.

**Conflicts of Interest:** The author declares no conflict of interest.

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
