# Peer review of "Experiences of Faith-Based Organizations as Key Stakeholders in Policy Responses to Human Trafficking"

_societies, doi:10.3390/soc13080193_

Round 1

Reviewer 1 Report

Experiences of Faith-Based Organizations as Key Stakeholders in Policy Responses to Human Trafficking

The authors are commended for taking on an in-depth and comprehensive qualitative study looking at the experiences of FBOs in human trafficking. The main concern with the article is the language used being leading or leaning and the need to expand the introduction and limitations. These concerns preclude the recommendation for publication at this time.

Introduction

·      Overall the introduction could be improved through expansion on the review of current literature. The authors mention repeatedly that FBOs play a large role in anti trafficking efforts, I do not dispute this but, for example, would appreciate an example from one of the cited articles that mentions with evidence their reach across the United States.

o   Similarly a review of how FBOs became so large in this sphere would be valuable to understand where this motivation came from, whether it started from missionary work or prostilitizing and how it’s evolved to the services offered today

·      The additions of headings would be helpful to guide the reader from one subject of the introduction to another

·      The language of the introduction does not remain neutral. The author is commended on presenting two sides of a sometimes-opposing story of FBOs and their role in human trafficking work. However, the language should be updated to reflect a non-leading tone. Here is a primary examples:

o   In their discussion of the “important roles” of FBOs at the beginning and halfway through page 3 of the introduction, the authors state the roles and things that FBOs do as facts “FBOs play important roles…FBOs can access…” however the wording in the case against FBOs is not stated as fact “FBOs have been accused of….It is also claimed that FBOs” this language needs to change either to state all as similar fact or add nuance to all of the current literature.

·      This statement needs a citation - FBOs are more likely to be single-issue anti-human trafficking organizations delivering direct service provision than their secular counterparts; they also make a significant contribution to political lobbying in the UK. (bottom of page 3)

·      It is recommended that this statement be removed and/or reworded as the current study does not examine the effectiveness of FBO’s trafficking-related services - There is still a lack of empirical research on the effectiveness of FBOs’ trafficking-related services. According to Barrows [8], “With such a high degree of engagement in the anti-trafficking movement, it is important to measure the impact of services provided by FBOs” (p. 288).

Methods

·      Please provide information on how many of each type of faith group you reached out to to assess recruitment scope.

Findings

·      It would be helpful to understand how many respondents said things that were then coded into each theme. The results continually report “Respondents said…” please add how many respondents said these things so the reader can get a sense of the prevalence.

Discussion

·      The first sentence is misleading. The fact that only 13 CFBOs participated does not immediately reveal they have “a substantial level of input” in human trafficking policy – please revise

·      Please avoid stating things as facts and instead point out that they are the results of the study throughout. For example, the sentence” The level of passion that many FBOs show in doing human trafficking work is challenging to understand, which often leads many people outside this circle to believe FBOs have hidden agendas to work in such a dangerous field of human trafficking.” Instead it is advised to rephrase these statements as presentation of the results (e.g. “Respondents reported that others find it challenging to understand the passion that they have in doing human trafficking work. They reported that this can often be seen by those not in FBOs as the organization having a hidden agenda.”)

o   This happens very frequently throughout the discussion

·      Please provide citation for

o   FBOs could be considered the leaders in shelter services, housing, and drop-in services for trafficking in the U.S.”

o   “Public grants have too many restrictions (i.e., time limits, invasive processes, too much paperwork, etc.) that can affect the quality of services for survivors.”

o   “possibly learning from FBOs’ distinctive competencies and approaches to advocating for policy change and providing various types of services for trafficking survivors.”

LImitations

Please include the limitation that the participants are all representatives of organizations they would like to be seen in a positive light and this likely shaped their answers as it would any representative.

Reviewer 2 Report

The 3 key research questions set out at p4 could be much clearer.  For example: 

(1) What faith motivates FBOs’ engagement in the field of human trafficking?

could become

(1) What role does faith play in motivating  FBO engagement in the field of human trafficking?

And so on ....

The research questions at p4 did not match the richness of the discussion and should be revisited. Eg there should be a question about the specific contributions of FBOs.

Ditto the conclusions were very brief and need to match the research questions.  It would also be interesting to have stronger observations on the particular \ distinctive capacities of FBOs 

p42 explain Maslow's Hierarchy of Needs

I am unclear why the author says at p47: The researcher’s direct involvement in all the stages of the research process may also have influenced the findings

There are some instances where the expression is awkward \ unclear and could be better expressed.  Eg:

p15 the sentence 'Daniel described four human trafficking policy advocacy accomplishments his FBO had' would be better as 'Daniel described four human trafficking policy advocacy accomplishments of his FBO.'

p19 'Trafficking survivors have too many needs that a single organization cannot address' would be better as 'Trafficking survivors have too many needs for a single organization to address'

There is some repetition eg p 10 and p11 the sentence: 'One of the FBOs could be considered one of its kind in addressing human trafficking from inside the corporate business system.'  Note the addition of 'in' 

The discussion around bottom of p27-top p28 is repetitive and needs editing

In fact the whole paper needs English editing and can be pruned down a little - it tends to be over-wordy. 

There is repetition of similar information re a federal liaison office pp35 and 46 but in the first reference there is a coy reference to a 'new administration' and at p46 to Biden's 

These are examples of a larger problem of language, expression and over-wordiness.  The script needs careful editing and attention to the structure of sentences. Often the meaning is obscured.  

For example, the sentence at p37: The fact that many FBOs relied more on uncorroborated stories than hard data to show their programs’ effectiveness was problematic because it hurt the impact of their work.

At p42: 'agencies liked meeting survivors where they were'

Reviewer 3 Report

This is an interesting and generally well-done piece of qualitative research on an important topic.  I was especially struck by the apparent paucity of research and evaluation on the effectiveness of the various services and programs.  The call for much more in this area is well-taken.
